# Emerging Topochemical Strategies for Designing Two-Dimensional Energy Materials

**DOI:** 10.3390/mi12080867

**Published:** 2021-07-23

**Authors:** Na Liu, Libo Chang, Jiang Wu, Jianhua Fang, Xu Xiao

**Affiliations:** 1Department of Petroleum, Oil and Lubricants, Army Logistic Academy of PLA, Chongqing 401331, China; liuna8911@163.com (N.L.); wujiang179@163.com (J.W.); 2Yangtze Delta Region Institute (Huzhou), University of Electronic Science and Technology of China, Huzhou 313001, China; changlibo@uestc.edu.cn; 3School of Electronic Science and Engineering, State Key Laboratory of Electronic Thin Film and Integrated Devices, University of Electronic Science and Technology of China, Chengdu 610054, China

**Keywords:** 2D materials, topochemical synthesis, MXene, energy applications

## Abstract

The unique properties of two-dimensional (2D) materials make them increasingly attractive in various fields, especially for energy harvesting, conversion, or storage. Simultaneously, numerous synthetic methods have been rapidly developed. Recently, topochemical strategies were demonstrated, and they show tremendous promising potential for synthesizing 2D materials due to their simplicity, scalability, and high efficiency. Considering the suitability of material structures and their synthesis methods, as well as the relationship between material properties and applications, it is necessary for researchers to comprehensively review and determine the prospects of 2D materials based on topological chemical synthesis methods and their related applications. Therefore, in this review, we systematically summarize and analyze the representative topochemical strategies for synthesizing 2D materials, including salt-templating methods for non-layered 2D materials, molten Lewis acid etching strategy for novel MXenes, and the chalcogen vapors etching and substituting strategy for phase-controlled 2D materials and so on, with the application of these 2D materials in energy-related fields including batteries, supercapacitors, and electrocatalysis. At the end of the paper, the corresponding perspective was also illustrated, and we expect that this could provide a reference for the future research in the field.

## 1. Introduction

Due to the lack of resource and the deteriorating environment, exploiting renewable energy has become one of the most urgent topics around the world. To date, a series of renewable energy such as wind energy, solar energy, and tidal energy have been developed [1,2,3,4,5,6]. However, given that the above energy sources are intermittent, it is necessary to find a way to supplement these sources; one approach is the utilization of energy storage technologies combined with energy conversion, which can provide an efficient use of renewable energy [7,8,9]. Two-dimensional (2D) materials show immense potential in the field of energy harvesting, conversion, and/or storage due to many outstanding physical and chemical performance, including high specific surface area and conductivity, abundant and active surface termination, processability, etc. [10,11,12]. However, the traditional synthesis methods of 2D materials have some limitations. For instance, vapor synthesis, including chemical or physical vapor deposition (CVD or PVD), can produce high-quality, large-size 2D materials on substrates. Although the volume of CVD or PVD synthetic materials may be sufficient for electronic devices, they are not sufficient for many other applications including energy storage [13,14,15]. In addition, 2D materials can be prepared by chemical synthesis or by stripping them in solution to obtain higher yield. However, the exfoliation is only applicable to layered materials with weak van der Waals forces that can be easily separated by mechanical cleavage or by intercalation agent-assisted cleavage [16]. Therefore, how to achieve high yield and high-quality synthesis of 2D materials is a huge challenge.

In recent years, topochemical synthesis emerged as a novel approach, whereby a novel 2D material can be synthesized by either substituting, adding, or extracting elements to or from precursors in an environment of liquid or gas. Moreover, the new obtained 2D material retains the structure or, at least, the morphology of the precursors [17]. For instance, the novel 2D transition metal nitrides, carbides, or carbonitrides (MXenes) were discovered by Yury Gogotsi and co-workers in 2011, which were obtained by etching the precursor (named MAX) with the removal of the A element, which is typical of topochemical strategy, where M is the transition metal element (such as Sc, Ti, Zr, Hf, V, Nb, Ta, Cr, Mo), A is third and/or fourth group element (such as Al and/or Si), and X is the C, N element and/or both [18,19]. The production of silicane and germanane, which are selectively etched from CaSi_2_ and CaGe compounds, respectively, also follow a similar strategy [20,21]. Some 2D zeolite can also be obtained by selectively etching 3D zeolites [22]. These methods, performed via selectively etching specific element in precursor by etchants, are commonly named as topochemical deintercalation, which greatly enriches the diversity of 2D materials.

A wet chemical solvothermal/hydrothermal treatment to molecular precursors is another effective method for synthesizing 2D materials. For instance, layered TMOs can be produced by self-assembly, with short chain alcohol cosurfactants and amphiphilic block copolymers confining the growth and stacking the oligomers (metal oxide precursor) along the specific direction [23]. Xiao et al. reported the synthesis of layered hexagonal-MoO_3_ (h-MoO_3_) via a salt-template method. In this process, the authors first prepared a molecular precursor solution and mixed it with a large volume of inorganic salt. Next, the 2D oxides were produced by annealing the mixture, which was obtained by drying the above solution. The targeted products manifested a large size of 100 μm and were easy to process to film. Subsequently, they succeeded in the transformation from 2D metal oxides (*h*-MoO_3_ and VO*_x_*) to 2D metal nitrides (MoN and V_2_N) following this strategy, by virtue of lattice matching of inorganic salt (as template) and targeted products [24,25]. These methods are called topochemical transformation, another topochemical strategy. This strategy is important for basic research because one can convert two materials according to their spatial structure and stoichiometric matching. For instance, a topochemical transformation from carbide MXenes to nitride MXenes, with improved electron conductivity, was achieved in this way in [26].

Undoubtedly, the emergence of the topochemical strategy provides a new idea for the synthesis of various novel 2D materials. Herein, this review summarizes the synthesis of some 2D materials including MXene, layered TMOs, TMDs, TMNs, and so on, based on the topochemical strategy with detailed mechanism, as well as their application in energy related fields as shown in Table 1. Finally, the prospective vista on future research on the topochemical strategy of 2D materials and their possible application scenarios are illustrated at the end of this paper.

## 2. The Synthesis of 2D Materials by Topochemical Deintercalation and Their Energy-Related Applications

Recently, a family of carbonitrides (MXenes) have rapidly drawn researchers’ attention due to the remarkable physical/chemical properties they have. They are pioneered by research groups led by Gogotsi and Barsoum in 2011 [19]. These materials have a formula of M_n+1_X_n_T*_x_*, where M is a transition metal (such as Ti, Sc, Zr, V, Hf, Nb, Ta, Mo, Cr, etc.), X represents carbon and/or nitrogen, and T*_x_* is the surface terminations (e.g., OH, O, F, etc.). In general, the preparation of MXene mainly involves the selective etching of the A layer of the MAX phase—their parent precursors by fluoride solution or fluorine containing molten salt, such as hydrofluoric acid (HF) aqueous solution, mixture of fluoride and hydrochloric acid or potassium fluoride/sodium fluoride/lithium fluoride mixed salt. The method of removing the atoms selectively and choosing suitable precursors and etchants is commonly referred to as a topochemical deintercalation synthesis, which can largely expand the family of 2D materials [17]. For example, the first MXene-Ti_3_C_2_ was achieved by etching the Al atomic layer in the MAX phase Ti_3_AlC_2_ by 50% concentrated HF solution [19]. In 2014, Ghidiu et al. synthesized a Li-intercalated MXene “clay” by the etchant of 5.0 M LiF and 6.0 M HCl [27]. Halim et al. etched the more active Ga atomic layer in the Mo_2_Ga_2_C MAX phase with the mixed solution of 12 M HCl and 3.0 M LiF, and they successfully prepared a layered Mo_2_CT_x_ [46]. It is obvious that the difference of bonding energy between the atomic layers is a key point for the above strategies [28,47].

Currently, more than 25 different MXenes have been synthesized by selective etching. They exhibited remarkable electrochemical performance, and attracted extensive attention in the field of energy applications. In 2013, Gogotsi and co-workers fabricated the MXene “paper” by the filtration of a Ti_3_C_2_T*_x_* solution. The MXene “paper” exhibited volumetric capacitances of 350 F cm^−3^ at 20 mV s^−1^ (and 450 F cm^−3^ at 2 mV s^−1^) in potassium hydroxide (KOH) electrolyte [29]. In 2014, the same group chose a mixed solution of HCl and LiF as the etchant and obtained Ti_3_C_2_ by etching Ti_3_AlC_2_ for 45 h at 40 °C [27]. The 5 μm freestanding Ti_3_C_2_ film performed good flexibility and a certain strength, which can be repeatedly folded with the intact structure remained. Moreover, conductive film or solid of any shape can also be fabricated. The MXene flake has a large lateral dimension and no significant nanoscale defects, indicating that the etchant consisting of HCl and LiF is more gentle than HF. Compared with previous work, it can be seen that Ti_3_C_2_T_x_ etched by HCl and LiF show an outstanding capacitive of 910 F cm^−3^ (246 F g^−1^) @2 mV s^−1^@1 M H_2_SO_4_, not only volumetrically but gravimetrically as well. Bak et al. synthesized 2D vanadium carbide MXene (denoted as V_2_CT*_x_*) by selectively etching the atomic layers of aluminum from V_2_AlC MAX phase with a 50% concentrated HF solution [28]. V_2_CT_x_ was studied as an anode material for Na-ion batteries, and it exhibits reversible charge storage with good cycling stability and high rate capability. The redox reaction of vanadium during Na^+^ intercalation/deintercalation implies that the transition metal (M) site in MXene is responsible for the electrochemical charge storage.

Not only can it be used as individual electrode materials, MXene can also cooperate with other materials to develop composites with enhanced electrochemical properties. Shao et al. obtained MXene (Ti_3_C_2_T_x_) and fabricated the composite of MXene (Ti_3_C_2_) onto polyester (PET) yarn via templated electrospinning. The PET@MXene yarn with great flexibility and strength was used as the electrodes of supercapacitor, and it showed high electrochemical performance (an ultra-high specific capacitance of 18.39 mF cm^−2^ at 5 mV s^−1^) as compared to graphene and CNT yarn supercapacitors (Figure 1a) [48]. Zhang et al. fabricated 3D porous Ti_3_C_2_T_x_/carbon nanotube (CNT) film (3D-PMCF) by a simple in-situ ice template strategy [49]. In their work, the 3D structure of Ti_3_C_2_T*_x_* significantly increased the exposed surface active sites and accelerates the ion transport, meanwhile maintaining good flexibility. Consequently, the flexible 3D-PMCF film delivered a capacitance of 375 F g^−1^ at 5 mV s^−1^ and retained 251.2 F g^−1^ at 1000 mV s^−1^ (Figure 1b) with good cycling stability, much superior to the conventional densely stacked Ti_3_C_2_T*_x_* film.

Hu et al. concluded that Ti_3_C_2_T_x_ MXene can be used as an anode with intercalation pseudocapacitive mechanism in H_2_SO_4_ solution to storage charge [50]. The features are as follows: an MX layer with high conductivity enables fast electron transport; a layered structure with open channels enables fast ion transport and intercalation; a transition metal with variable valence enables charge transfer; and the -O functional groups on the transition metal surface provides redox active sites, which is consistent with the characteristic of intercalation pseudocapacitance.

In order to be removed from the dependence of the current preparation strategy on toxic fluoride and in turn develop more etchable MAX phase precursors whose A-site is not the conventional Al atomic layer, great efforts have been made in studying the development of new precursors, the influence of novel etching methods, as well as controllable synthesis and potential applications. In 2019, Huang et al. found that the Ti_3_AlC_2_ MAX phase had significant reaction in molten ZnCl_2_ (Figure 2a) [50]. The Zn element replaces Al element to form a new MAX phase-Ti_3_ZnC_2_ at the initial step. The small angle shift of the peaks in XRD pattern (Figure 2b) at (103), (104), (105), and (000l) indicates the layer spacing is expanded, which could be explained by the larger atomic radius of Zn. The SEM images of the MAX phase before and after the substitution (Figure 2c,d) reveal that the material basically maintains the original layered structure. Atom-resolved scanning transmission electron microscopy (STEM)/energy dispersive spectrometer (EDS) images in Figure 2e,f show the atomic thickness of the Zn layer in Ti_3_ZnC_2_. STEM mapping in Figure 2g further shows that the Zn layer is evenly embedded between the Ti layer and C layer. When employing excess ZnCl_2_, the process is highly similar to the etching of Ti_3_AlC_2_ by HF solution, in which Zn^2+^ plays the role of H^+^ in HF solution, while Cl^-^ is equivalent to F^-^ and finally coordinates with M atoms. 

Figure 2h shows the SEM image of Ti_3_C_2_Cl_2_, demonstrating the typical accordion structure of MXene. The XRD pattern in Figure 2i also reveals that the crystal structure of MXene prepared by molten ZnCl_2_ is more ordered than that of samples etched by HF solution or mixed molten salt. Moreover, the HR-TEM image indicates that the Cl atom terminated on the surface of Ti_3_C_2_ (Figure 2j). This work not only obtained a series of M_n+1_ZnX*_n_* MAX phases through the substitution of the A-element, but it also realized new MXenes with Cl termination.

The same group further expanded the etching strategy to a wider MAX phase family (e.g., A = Al, Zn, Si, Ga, etc.) by using a variety of Lewis acid chloride molten salts (e.g., ZnCl_2_, FeCl_2_, CuCl_2_, AgCl, etc.) [31]. More importantly, they successfully constructed a Gibbs free energy map of redox potential/displacement reaction between the cation and the A-element in a high-temperature molten salt environment, which provides a general guidance for synthesizing 2D MXene by etching the MAX phase with Lewis acid molten salt. For instance, they carried out a detailed analysis using Ti_3_SiC_2_ MAX phase. Ti-Si bonding in Ti_3_SiC_2_ is stronger than Ti-Al in Ti_3_AlC_2_. Considering that the redox potential of Cu/Cu^2+^ is −0.43 eV while that of Si/Si^4+^ is −1.38 eV, the Cu^2+^ ions in molten salt can easily oxidize the Si atom to Si^4+^, followed by Si^4+^ and Cl^-^ forming SiCl_4_ gas and then escaping from the sublayer of Ti_3_C_2_. At the same time, Cu^2+^ is reduced to Cu. The residual Cu in the product can be removed by an ammonium persulfate (APS) solution and other methods. Finally, Ti_3_C_2_T*_x_* (T*_x_* = Cl, O) MXene with mixed functional groups of Cl and O can be obtained. According to the electrochemical redox potential of the A-site element and Lewis chloride acid salt, the electronic ability of the A-site element from Lewis acid salt can also be evaluated, and thus suitable Lewis acid molten salt and MAX phase could be chosen in advance. Ti_2_AlC, Ti_3_AlC_2_, Ti_3_AlCN, Nb_2_AlC, Ta_2_AlC, Ti_2_ZnC, Ti_3_ZnC_2_, and other MAX phases could be etched by appropriate chloride molten salts (e.g., CdCl_2_, FeCl_2_, CoCl_2_, CuCl_2_, AgCl, NiCl_2_) to obtain their corresponding (e.g., Ti_2_CT*_x_*, Ti_3_CNT*_x_*, Nb_2_CT*_x_*, Ta_2_CT*_x_*, Ti_2_CT*_x_*, Ti_3_C_2_T*_x_*) MXenes. Different from the current liquid phase etching, the synthesis of MXenes by the Lewis acid molten salt method greatly improves the chemical safety of the experimental process and reduces the difficulty and the cost of waste disposal, which is expected to further accelerate the research of MXene in various fields. A negative electrode of Ti_3_C_2_ MXene material obtained through this molten salt synthesis method delivers a Li^+^ storage capacity of up to 738 C g^−1^ (205 mA h g^−1^) with a high charge–discharge rate and a pseudocapacitive-like electrochemical signature in a 1 M LiPF_6_ carbonate-based electrolyte. MXenes prepared via this molten salt synthesis route may be suitable for high-rate negative-electrode materials for electrochemical energy storage applications. The combination of a mirror-like electrochemical signature during Li^+^ intercalation/deintercalation in nonaqueous Li^+^-containing electrolyte, together with a high capacity, high-rate discharge and charge performance (<1 min), and the low operating potential range (0.2–2.2 V versus Li^+^/Li), makes these MXenes prepared via a molten salt route potentially useful as negative electrodes in electrochemical energy storage devices (batteries and Li-ion capacitors).

Very recently, Talapin et al. introduced a similar strategy to introduce and remove surface groups by performing replacement and elimination reactions in molten inorganic salts (e.g., CdCl_2_, CdCe_2_) [51]. In their work, a series of MXenes with O, NH, S, Cl, Se, Br, and Te surface terminations were successfully prepared with unique structures, including the surface groups that controlled interatomic distances’ lattice and giant (e.g., >18%) in-plane lattice expansion. It was concluded that the surface groups were not spectators but were active contributors to the MXene superconductivity, and the chemical bonds inside an extended MXene stack can be rationally designed in a way that is more typical for molecular compounds.

## 3. The Synthesis of 2D Materials by Topochemical Transformation and Their Energy-Related Applications

### 3.1. Two-Dimensional TMOs

Two-dimensional TMOs have demonstrated promising prospect in energy harvesting/storage system, sensing, detection [52,53,54,55,56], due to their superior electrochemical stability, high theoretical capacities, atomic thickness, etc. Currently, the synthesis of 2D TMOs mainly focuses on the natural layered precursors such as *α*-MoO_3_ and MnO_2_ [34,35], and there are few studies on non-layered TMOs. However, 2D non-layered TMOs may have unique advantages over their layered counterparts in terms of practical applications. For instance, *h*-MoO_3_ possesses a larger inner crystal channel (e.g., 6.27 Å) compared to *α*-MoO_3_, which is favorable for the rapid ion transport and is expected to show superior performance in energy storage devices. However, the synthesis of 2D TMOs with non-layered structures, such as 2D *h*-MoO_3_, is not available through the traditional exfoliation methods because of the instinct bulk structure of the precursor. Thus, the exploration of effective routes to the synthesis of 2D TMOs is challenging and significant.

Xiao et al. developed a general salt-templating method for the synthesis of a variety of non-layered 2D TMOs, which combined the merits of the high quality of gas phase synthesis and the high yield of liquid phase synthesis [36]. The key point of this synthesis strategy is the high lattice matching between the target product and the salt template (e.g., NaCl, KCl). As shown in Figure 3a, the precursor was completely dissolved in the ethanol solution, followed by an even mixing with a large amount of salts. After drying, the salt template is tightly and evenly wrapped by the precursor with thickness of ~1 nm. Finally, the target TMOs can be obtained by calcination under Ar atmosphere and by dissolving the salt template in water. It is worth mentioning that the thickness of the as-prepared TMOs can be easily adjusted by changing the dosage ratio of the precursor to the salt template.

To reveal the mechanism of the salt-templating growth of 2D TMOs, the synthesis of 2D MnO on KCl was analyzed in detail. Figure 3b shows the lattice matching diagram of the MnO (001) plane placed on the (001) plane of KCl, whose lattice parameters (a_1_ and a_2_) are 0.4442 nm and 0.3138 nm, respectively. Considering that the crystal structure of KCl is symmetrical, its diagonal √2 a_2_ is 0.4437 nm, which is very close to the crystal lattice constant of MnO. In other words, the crystal lattice of KCl overlaps with that of MnO when the crystal plane of MnO is rotated by 45°. As shown in Figure 3c, the X-ray diffraction (XRD) peaks of KCl and of the 2D MnO mixture (before washing) indicate that the (200) and (2$\overset{-}{2}$0) crystal planes of MnO are very close to the (1$\overset{-}{1}$0) and (200) crystal planes of KCl. In addition, the calculation results in Figure 3d further demonstrate that the lattice mismatch degree between KCl and MnO is only about 0.11 %. To verify that the ultra-thin 2D structure is attributed to the high lattice matching, Na_2_SO_4_ was also utilized as a salt template. The calculation results (Figure 3e) demonstrate that the lattice mismatching degree in different directions is relatively high, and the experimental results further confirm that the product would be particles rather than nanosheets. The morphology and structure of the obtained 2D MnO were carefully characterized. The scanning electron microscope (SEM) image shows that the lateral size of the nanosheet is about 20 μm (Figure 3f), which is significantly superior to that synthesized by liquid phase exfoliation and the wet chemical strategy, which gives the material great application prospect in microelectronic devices. The atomic force microscopy (AFM) image (Figure 3h) indicates that the average thickness of the nanosheet is about 1.06 nm. Furthermore, this strategy was successfully extended to other type of 2D non-layered TMOs such as hexagonal-MoO_3_ (*h*-MoO_3_), MoO_2_, and hexagonal-WO_3_ (*h*-WO_3_).

The two-dimensional *h*-MoO_3_ was selected for exploring electrochemical performance, due to its high theoretical capacity (1.5 Li/Mo). With atomically thin 2D oxides, reversible and fast surface reactions were achieved, compared to bulk MoO_3_. Because the ions can easily diffuse from the electrolyte into the restacked film to react with the fully exposed surface-active sites of the 2D oxides, 2D *h*-MoO_3_ achieved high capacitance at high rates in pseudocapacitor applications.

In their work, Xiao et al. also explored the multilayer electrodes of *h*-MoO_3_ by a three-electrode configuration in six different aqueous electrolytes (H_2_SO_4_, Al_2_(SO_4_)_3_, MgCl_2_, NaCl, KCl). As shown in Figure 3i, the highest volumetric capacitance of 600 F cm^−3^ was achieved in H_2_SO_4_ at 5 mV s^−1^, which is larger than chemically converted graphene in H_2_SO_4_ (256 F cm ^−3^) [37], and a volumetric capacitance of 300 F cm^−3^ was achieved in Al_2_(SO_4_)_3_ electrolyte at 5 mV s^−1^. The surprising results indicated that the 2D oxides can also be used in Al-ion batteries. Owing to the narrow potential window of aqueous electrolytes, Xiao et al. chose LiClO_4_ solution with a 1:1 mixture of ethylene carbonate and dimethyl carbonate as the organic electrolyte to maximize the energy stored in 2D *h*-MoO_3_. As shown in Figure 3j, a gravimetric capacity of 996 C g^−1^ (277 mAh g^−1^) of *h*-MoO_3_ at 2 mV s^−1^ was obtained, which was much larger than α-MoO_3_ in 1 M LiClO_4_ with propylene carbonate (~580 C g^−1^, 161 mAh g^−1^) previously reported [38].

### 3.2. Two-Dimensional TMDs

The preparation of 2D materials with higher yield and tunable structure is of importance but remains challenging. Yang et al. proposed a novel strategy for synthesizing 2D materials in 2020 [39], through which various ultra-stable 2D TMDs with a high monolayer rate can be prepared scalable. They treated the MAX phase with the gas or vapor of chalcogenide H_y_Z (y = 0, 2; Z = S, Se, Te). Due to the stronger covalent M–X bonding than the metallic M–A bonding, the A layer in non-van der Waals solid precursor would react with the H_y_Z gas to form a gaseous product (AZ) with high vapor pressure, which could be separated from the reaction system under a high temperature (i.e., 873–1373 K). In the meantime, the M layer was transformed into a 2D TMDs (MZ).

This work reveals that the topological transformation can be controlled by the enthalpy and vapor pressure. By regulating the composition of the MAX phase (ternary or quaternary MAX phase) and introducing a third reactant (e.g., phosphorus vapors) into the reaction system, the 2D transition metal sulfides (2H/1T phase) with high monolayer yield (≈30%) and ultra-high temperature stability (>1000 °C) can be synthesized. When layered TMCs (e.g., Mo_2_C) is adopted as the reactant, MoS_2_ with ultra-high monolayer rate (≈90%) can also be achieved. The researchers synthesized 14 TMDs with controllable phase structure and an ultra-high monolayer rate through this topological transformation, including 7 TMDs (i.e., Ti_5_S_8_, TiSe_2_, NbS_2_, NbSe_2_, MoS_2_, MoSe_2_, TaS_2_), 5 doped TMDs (i.e., Y-doped WS_2_, Nb-doped TiSe_2_, Y/P-doped WS_2_, P-doped MoS_2_, P-doped TiSe_2_), as well as a composite MoSe_2_/TiSe_2_. This method can directly transform non-van der Waals solid materials into single-layer TMDs, which are universal and suitable for industrial production. This method is expected to synthesize a series of van der Waals 2D materials with ultra-high monolayer yield and high temperature stability on a high yield, enjoying broad application prospects in energy storage/conversion, microelectronic devices, and other fields.

Voiry et al. studied the catalytic activity of the two kinds of coordination molybdenum atoms for MoS_2_—octahedral (1T phase) and trigonal prismatic (2H and 3R phases) for hydrogen evolution reaction (HER) [40]. The highly conducting 1T phase MoS_2_ nanosheets exhibited superior catalytic activity toward the evolution of hydrogen with a notably low Tafel slope of 40 mV/dec after removing the excess of negative charges from the surface of the nanosheets. On the other hand, the activity of 2 H MoS_2_ was significantly reduced after oxidation and 1 T MoS_2_ remained unaffected, which indicates that the edges of the nanosheets are not the main active sites and that the electrical conductivity is important for catalytic activity.

### 3.3. Two-Dimensional TMXs (X = Nitrides, Carbides, Phosphides)

The topochemical transformation strategy presented above shows strong universality, by which 2D TMOs can be realized through the lattice matching of the salt template and the target oxides. Furthermore, it was also found that by substituting the oxygen element in the 2D TMOs with other elements, a variety of non-layered 2D TMXs—including nitrides, carbides, and phosphides—could also be synthesized on a large scale.

#### 3.3.1. Nitrogen Substitution

As a new member of 2D materials family, 2D TMNs, such as V_2_N, MoN, W_2_N have been widely studied in energy storage, catalysis, optoelectronic devices, and ferromagnetism due to their unique physical and remarkable chemical properties [26,41,57,58,59,60]. Specially, there is evidence showing that the increase of nitrogen content in 2D TMNs could result in an elevated oxidation state and in a regulated band gap, leading to the leap of certain properties [61,62,63,64]. At present, the preparation of 2D TMNs mainly depends on the CVD-based, high-temperature gas synthesis, as well as the etching of very few MAX phase precursors (e.g., Ti_4_N_3_) [26,41,57,58]. The former is more dependent on the equipment and the process, while the latter requires a nitrogen-containing MAX precursor, which is less achievable and not suitable for the non-layered 2D TMNs. Thus, the development of efficient and high-yield strategies, especially those applicable towards non-layered 2D TMNs, remains challenging.

Xiao et al. successfully realized a scalable synthesis of 2D TMNs by substituting the O element in 2D TMOs with an N element [25]. The schematic illustration of the synthesis of 2D MoN is shown in Figure 4a. Here, 2D hexagonal-MoO_3_ nanosheets produced on the surface of the salt template is further ammoniated in an NH_3_/Ar atmosphere. The 2D MoN can be obtained after washing and purifying. The TEM image (Figure 4b) shows overlapped single-layered 2D MoN nanosheets. The high-resolution TEM (HR-TEM) and selected area electron diffraction (SAED) images (Figure 4c) further reveal that 2D MoN is single-crystalline with a hexagonal structure. In addition, from the HR-TEM image of the flake edge (Figure 4d), the thickness of the 2D MoN is estimated to be around 0.71 nm. Figure 4e,f shows the side and top views of the monolayer MoN, respectively, from which it could be observed that the thickness of 2D MoN is around 0.7 nm. The density of state of 2D MoN in Figure 4g shows the high density of state at the Fermi level, indicating that the material is metallic and mainly attributed to the *d* orbital electrons of Mo atom. Moreover, 2D W_2_N, V_2_N, Mn_3_N_2_, and nitrogen-rich Mo_5_N_6_ and hexagonal-W_2_N_3_ were also prepared by utilizing this strategy [25,41,42,43]. It is worth mentioning that Xiao et al. adopted an anhydrous NH_3_ atmosphere to substitute the NH_3_/Ar mixed gas atmosphere, and they prepared a series of 2D arrays of TMNs nanocrystals such as CrN, NbN, TiN [43]. As a demonstration of their application in a lithium–sulfur battery (Figure 4h,i), it was shown that 2D arrays of several nitrides can achieve a high initial capacity of >1000 mAh g^−1^ at 0.2 C and only about 13% degradation over 1000 cycles at 1 C under a high areal sulfur loading (>5 mg cm^−2^). Different from the single-crystal materials mentioned above, the employment of anhydrous NH_3_ leads to the partial etching and atom substitution of the 2D nanosheet at elevated temperature, and this eventually obtained an exotic structure of local nanocrystal and overall 2D morphology.

Jin et al. utilized a Ni-inducing growth method to design and synthesize Mo_5_N_6_, which are atomically thin nitrogen-rich nanosheets [44]. In these cases, the transition metal component and salt crystal serve as the catalyst and template that induce the formation of anomalous metal nitrides. They speculated that the same theory is suitable for synthesizing other nitrogen-rich TMNs. The as-prepared 2D Mo_5_N_6_ nanosheets show enhanced HER activity compared to conventional nitrogen-deficient TMNs in various electrolytes (pH 0−14). Furthermore, the high HER activity and high stability of the two-dimensional Mo_5_N_6_ nanosheets in natural seawater were discovered, which were superior to other TMNs and even the Pt benchmark.

#### 3.3.2. Carbon Substitution

The salt-templating method was also utilized for the synthesis of 2D TMC and composites. In 2018, Xiong et al. developed a 2D Mo/Mo_2_C heterostructure [65]. Briefly, the generated 2D MoO_3_ on the surface of NaCl was calcined in CH_4_/H_2_ atmosphere (Figure 5a), leading to the reduction and substitution of O atom in 2D MoO_3_. The heterostructure started to appear when the temperature reached 650 °C, as shown in the XRD pattern (Figure 5b). The typical layered structure of Mo/Mo_2_C heteronanosheets (Mo/Mo_2_C-HNS-750) can also be observed in the HR-TEM image (Figure 5c,d), showing that the two lattice spacings (e.g., *d* = 1.57 Å; *d* = 2.37 Å) correspond to the (200) and (002) crystal faces of Mo and *β*-Mo_2_C, respectively. As depicted in Figure 5e, the corresponding energy dispersive spectroscopy (EDS) analysis further indicates the staggered combination of Mo and *β*-Mo_2_C on nanoscale. This unique 2D structure enables Mo and *β*-Mo_2_C to form an abundant double-phase interface (DPI) composed of reactive catalytic sites (*β*-Mo_2_C edges) and charge transfer channels (Mo), which greatly promotes the performance in catalysis. Therefore, HER electrocatalytic performance with an overpotential of merely 89 mV to drive a current density of 10 mA/cm^2^ in 0.5 M H_2_SO_4_ was achieved. Moreover, the Mo/Mo_2_C-HNS demonstrated stable chronoamperometric electrolysis for 20 h and an impressive cycling stability with negligible overpotential decay over 4000 sweeps. These findings demonstrated Mo/Mo_2_C-HNS as a potential metal-free electrocatalyst toward the HER.

#### 3.3.3. Phosphorus Substitution

Two-dimensional TMPs have been widely investigated in diverse fields including energy storage, electrocatalysis, and ferromagnetism because of the unusual physics and properties in the 2D confines [66,67,68,69], whereas the strong covalent bonds in bulk TMPs restrict the synthesis of 2D TMPs via conventional etching and exfoliation methods. Currently, wet chemical synthesis is the widely used method for obtaining 2D TMPs [70,71], even though this strategy still needs to be further improved in terms of the lateral size, thickness, and the yield of the target product.

The success of substituting oxygen in 2D TMOs by nitrogen and carbon for preparing 2D TMNs and TMCs inspired researchers to further extend this strategy to 2D TMPs [45]. As shown in Figure 6a, metal nitrate precursors and (NH_4_)_2_HPO_4_ are coated uniformly on the surface of a salt template (e.g., NaCl and KCl*)*, and then the hybrid composites are annealed in H_2_/Ar atmosphere to form 2D TMPs. The obtained dispersions of Co_2_P, MoP_2_, Ni_12_P_5_, and WP_2_ are displayed in Figure 6b. Figure 6c shows the nanosheet morphology of the 2D Co_2_P. The lateral size of the nanosheet is about 4 μm. The SAED image inserted in Figure 6d shows two crystal planes, i.e., (001) and (210), with an angle of 90°, from which the crystal axis of Co_2_P can be determined as ($\overset{-}{1}$20). The lattice spacing fringes (i.e., 0.351 nm, 0.261 nm) in Figure 6e correspond to the spacing of the (001) and (210) crystal faces in the SAED, respectively. The phase structure is further determined by XRD analysis (Figure 6f), and the diffraction peaks are attributed to the orthogonal crystal phase of Co_2_P. X-ray photoelectron spectroscopy (XPS) in Figure 6g reveals the generation of Co–P bonding, and the XRD pattern of Co_2_P @ KCl mixture in Figure 6h indicates that the crystal lattice of 2D Co_2_P could perfectly match that of KCl, validating the lattice-matching mechanism of the salt-templating method.

Among the prepared 2D TMPs, Co_2_P is well-suited for HER. Due to the weak electronegativity of phosphorus, the Co–P bond has a high electron density. Therefore, the researchers compared Co_2_P nanoparticles—which were synthesized in the same way without salt as templates—with 2D Co_2_P. As shown in Figure 6i, 2D Co_2_P shows a very low overpotential of 13 mV versus the reversible hydrogen electrode (RHE) in a 0.5 M H_2_SO4 solution electrolyte, which was much lower than that of Co_2_P nanoparticles (30 mV), and an overpotential of 41 mV at a current density of 10 mA cm^−2^ as well as good stability. It is believed that the family of 2D materials can be expanded due to the existence of 2D TMPs. Furthermore, a new avenue could be opened for designing exposed catalysts, and this provides possibilities for the desired functionality.

## 4. Conclusions and Perspectives

In this review, we summarized the topochemical synthetic methods of 2D materials, including topochemical deintercalations (such as that from MAX to MXene, from CaSi_2_ to germanane by etching the A element, Ca element in solution), as well as topochemical transformations (such as from MoO_3_ to Mo_2_C and/or MoN with a salt as template). The obtained 2D materials via the strategies show extensive application and outstanding performance in energy-related fields. There is no doubt that the emergence and development of topochemical strategies for 2D materials would break the limitation of traditional methods and raise a tendency of exploring new materials and specific properties, supplementing the family of 2D materials. Furthermore, numerous fields are expected to be adapted by 2D materials synthesized via the strategies.

Over the next decades, we believe that the research directions should focus on following steps. First, the utilization of theoretic calculation can reduce the number of explorations and the difficulty of the process for researchers by predicting the existed possibility of precursors and the targeted products via theoretic simulation, which could vastly minimize the experiment cycles and improve the experiment efficiency. The second is to exploit novel approaches with the superior performance of 2D materials. Although the 2D materials have achieved ground-breaking progress, the balance among theoretic performance, actual expression, and practical application is still not easy to be grasped. Therefore, the contradiction may be mitigated with the emergence of some new materials. Finally, with the development of society, especially with information technology, which is deeply changing lifestyles and ways of production, the diversities and multifunctionality of future materials and devices are expected to become urgent. Therefore, researchers should not only exploit new materials but also explore their various possible applications in combination with actual social needs.

## Figures and Tables

**Figure 1 micromachines-12-00867-f001:**
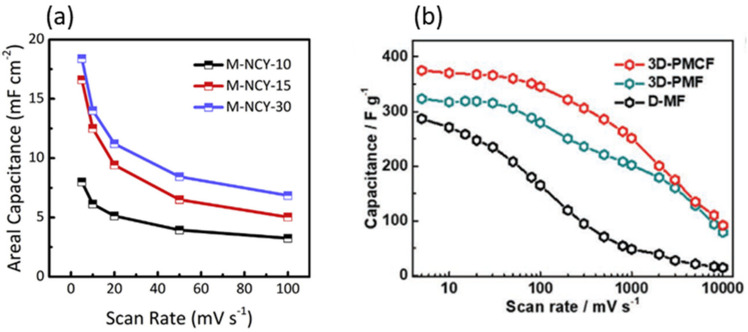
(**a**) Electrochemical performance of MXene-based nanofibers coated yarns (NCY) devices (M-NCY-10, M-NCY-15, and M-NCY-30 referred to the yarns obtained after 10, 15, and 30 min of electrospinning respectively)—variation in areal capacitances with scan rates. Reprinted (adapted) from Reference [48], copyright (2018) Elsevier. (**b**) Plots of gravimetric capacitance of 3D-PMCF and other counterparts against different scan rates. Reprinted (adapted) from Reference [49], copyright (2020) John Wiley and Sons.

**Figure 2 micromachines-12-00867-f002:**
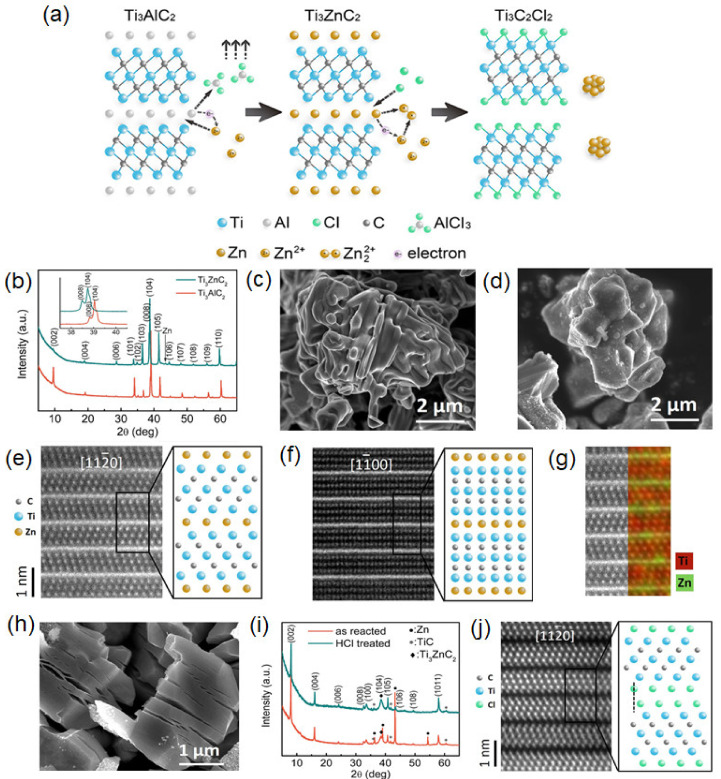
(**a**) Schematic illustration of the preparation of Ti_3_ZnC_2_ and Ti_3_C_2_Cl_2_. (**b**) XRD patterns of Ti_3_AlC_2_ and Ti_3_ZnC_2_. (**c**) SEM image of Ti_3_AlC_2_. (**d**) SEM image of Ti_3_ZnC_2_. (**e**,**f**) HR-STEM image of Ti_3_ZnC_2_. (**g**) HR-STEM and the corresponding EDS map of Ti_3_ZnC_2_. (**h**) XRD patterns from the as-reacted product and HCl-treated product. (**i**) SEM image Ti_3_C_2_Cl_2_ in the as-reacted sample. (**j**) HR-STEM image of Ti_3_C_2_Cl_2_. Reprinted (adapted) from Reference [50], copyright (2019) American Chemical Society.

**Figure 3 micromachines-12-00867-f003:**
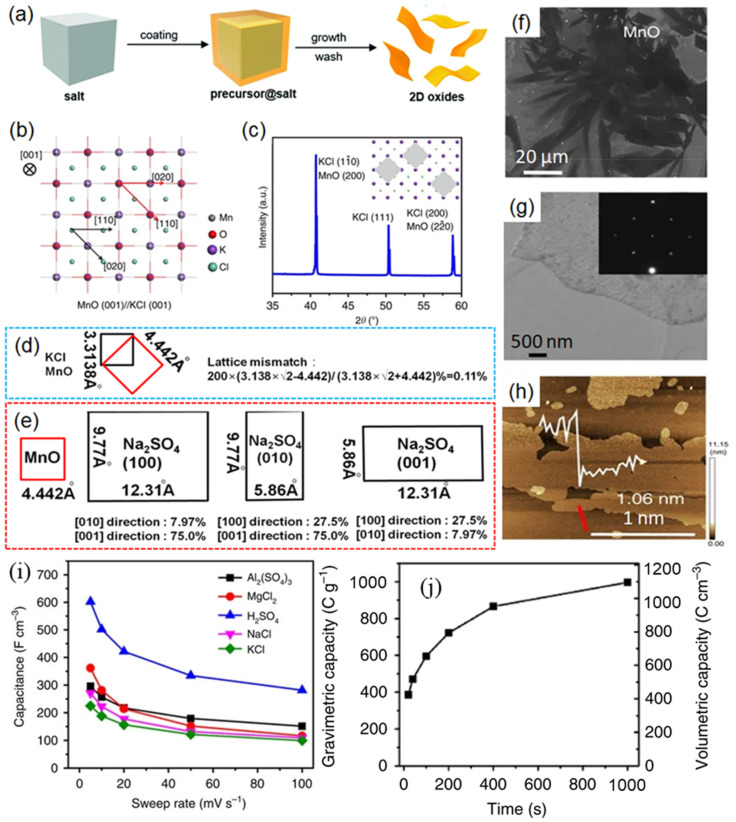
(**a**) Schematic illustration of salt-templating synthesis. (**b**) Proposed mechanism of the salt-templating growth of 2D MnO. (**c**) XRD pattern of 2D MnO on KCl. The inset is a schematic representation of MnO on KCl. (**d**) Lattice mismatch of MnO and KCl. (**e**) Lattice mismatch of MnO and Na_2_SO_4_. (**f**) SEM, (**g**) TEM, and (**h**) AFM images of 2D MnO. The inset of (**g**) shows the corresponding selected area electron diffraction (SAED) pattern. The inset of (**h**) shows the corresponding thickness of the 2D oxide. Scan line is shown in red. (**i**) Electrochemical performance of 2D *h*-MoO_3_ in aqueous electrolytes: volumetric capacitance as a function of the sweep rate in different aqueous electrolytes. (**j**) Electrochemical performance of 2D *h*-MoO_3_ in a Li-ion-containing organic electrolyte: gravimetric and volumetric capacity as a function of charging/discharging time. Reprinted (adapted) from Reference [36] with permission from Springer Nature.

**Figure 4 micromachines-12-00867-f004:**
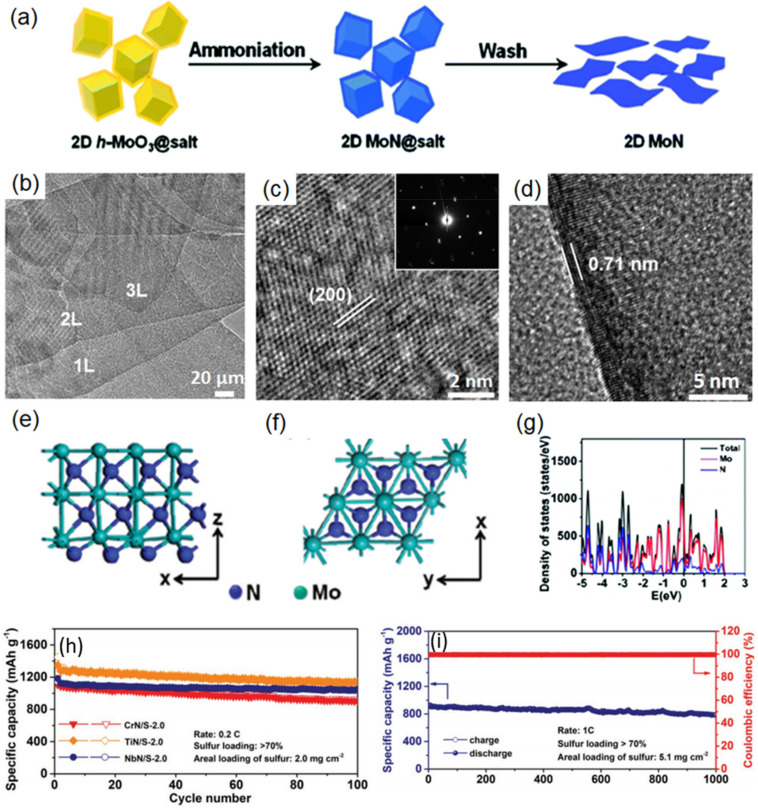
(**a**) Schematic illustration of 2D MoN synthesis. Side and top view images (**b**,**c**) of the atomic structure of 2D MoN. (**d**) Density of states of 2D MoN. (**e**) TEM image of overlapped single-layer MoN nanosheets. (**f**) HR-TEM shows that 2D MoN is single-crystalline with a hexagonal structure. The inset is a SAED that demonstrates the single-crystal structure of 2D MoN. (**g**) Electron density of states (DOS) of 2D MoN. No bandgap was found in 2D MoN, implying its metallic character. Reprinted (adapted) from Reference [25], copyright (2017) American Chemical Society. (**h**) Cycling performance of Li–S cells with CrN/S, TiN/S, and NbN/S cathodes at 0.2 C for 100 cycles (1 C = 1674 mA g^−1^). (**i**) Cycling performance of NbN/S-5.1 cathode at 1 C for 1000 cycles. Reprinted (adapted) from Reference [43], copyright (2019) John Wiley and Sons.

**Figure 5 micromachines-12-00867-f005:**
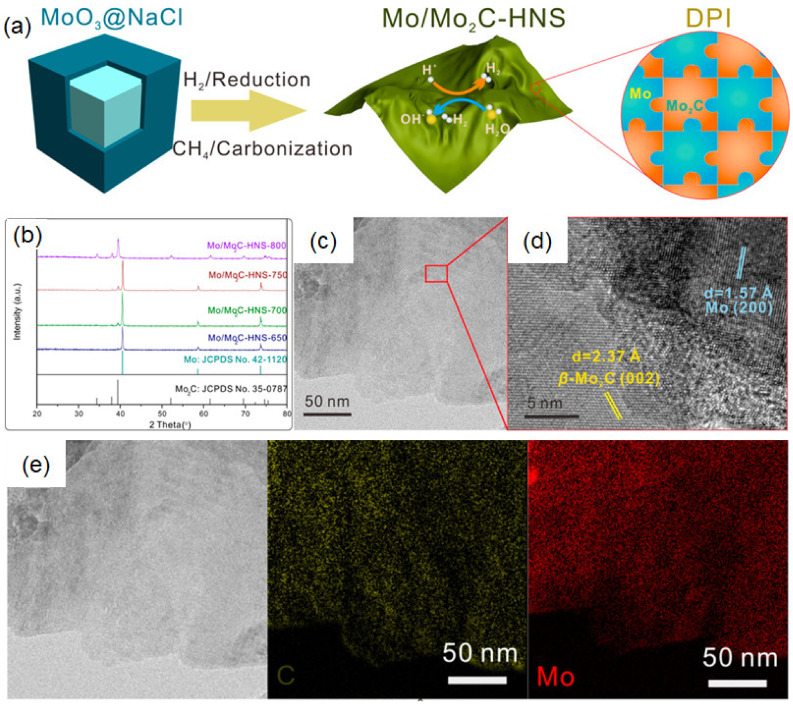
(**a**) Schematic illustration of the synthesis progress of Mo/Mo_2_C-HNS. Characterization of Mo/Mo_2_C-HNS-750: (**b**) the XRD patterns; (**c**) TEM image, and (**d**) HR-TEM image. (**e**) EDS elemental mappings. Reprinted (adapted) from Reference [65], copyright (2018) American Chemical Society.

**Figure 6 micromachines-12-00867-f006:**
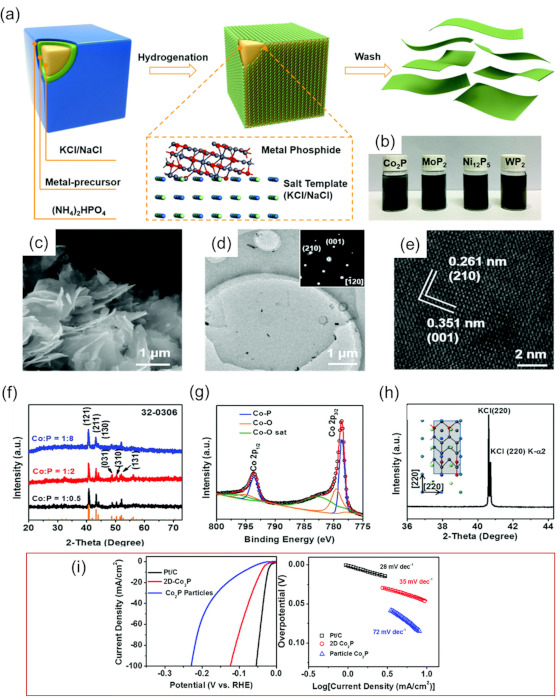
The characterization and synthesis mechanism of 2D Co_2_P. (**a**) Schematic diagram of the preparation of 2D metal phosphides. (**b**) The optical images of the as-synthesized 2D metal phosphide dispersions. (**c**) SEM, (**d**) TEM, and (**e**) HR-TEM images of 2D Co_2_P (inset: the corresponding SAED pattern). (**f**) XRD patterns of Co_2_P with different atomic ratios in the precursor. (**g**) XPS spectra of Co_2_P. (**h**) XRD pattern of 2D Co_2_P on KCl. (**i**) Polarization curves and Tafel plots of 2D Co_2_P, Co_2_P particles, and commercial Pt/C. Reprinted (adapted) from Reference [45], copyright (2018) Royal Society of Chemistry.

**Table 1 micromachines-12-00867-t001:** A summary of 2D materials produced by topochemical strategies and their application introduced in energy-related field of this manuscript.

Topochemical Strategies	Materials Categories		Energy-Related Applications	References
Topochemical deintercalation	MXenes	Ti_3_C_2_T*_x_*, V_2_CT*_x_*, Mo_2_CT*_x_*, Ti_2_CT*_x_*, Ti_3_CNT*_x_*, Nb_2_CT*_x_*, Ta_2_CT*_x_*, Ti_2_CT*_x_*, etc.	Electrocatalysts;	[27,28,29,30,31]
			supercapacitors;	
			batteries	
	Germanane		Photocatalysis	[32]
	2D zeolite		Electrocatalysis	[22,33]
Topochemical transformation	2D TMOs	α-MoO_3_, *h*-MoO_3_, MnO, MnO_2_, MoO_2_, *h*-WO_3_, etc.	Photocatalysis; electrocatalysts; supercapacitors; batteries	[34,35,36,37,38]
	2D TMDs	Ti_5_S_8_, TiSe_2_, NbS_2_, NbSe_2_, MoS_2_, MoSe_2_, Y-doped WS_2_, P-doped MoS_2_, Y/P-doped WS_2,_ TaS_2_, etc.		[39,40]
	2D TMNs	V_2_N, MoN, W_2_N, CrN, NbN, TiN, Mo_5_N_6_, *h*-W_2_N_3_, etc.		[25,41,42,43,44]
	2D TMPs	Co_2_P, MoP_2_, Ni_12_P_5_, WP_2_, etc.		[45]

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
