# Peer review of "Emerging Topochemical Strategies for Designing Two-Dimensional Energy Materials"

_micromachines, 2021, doi:10.3390/mi12080867_

Round 1

Reviewer 1 Report

1) information of the authors have different font sizes.

2) Line 113 - The Celsius abbreviation has different a different font from the rest of the text

3) None of the Figures have the reference

4) If possible, try to increase the resolution of the images

Reviewer 2 Report

This paper reviewed the topochemical strategies for synthesizing 2D materials and the energy applications based on these materials. The introduction was well written, although the other sections were written like a summary. Overall, this paper is interesting, and it consists of many related references. I have several suggestions, which may help to improve the article.

1) By using iThenticate, it is a 9% overlap between this paper and a previous paper, "Topochemical synthesis of 2D materials" [Chem. Soc. Rev., 2018,47, 8744-8765]. Therefore, the authors should consider NOT using the same sentence, such as "broadly defined as adding, extracting or substituting elements to or from precursors in an environment of liquid or gas for synthesis of new materials which retain the structure or, at least, morphology of precursors."

2) For all figures, the authors have to add the statement for permission; for example, "Adapted with permission from Ref. [xxx]".

3) The authors should avoid using adjectives in a scientific article. For example, "excellent" was used up to 6 times in the article.

3) The title of Sec. 2 at line 86 and Sec. 3 at line 221 are the same.

4) "3.12. D TMOs" at line 222 should be "3.1. 2D TMOs". 

5) "3.12. D TMDs" at line 298 should be "3.2. 2D TMDs".

6) "3.32. D TMXs" at line 343 should be "3.3. 2D TMXs", etc. Could you please check the title of each section carefully?

7) If the authors show the symbols in the figure, the authors should define the meaning of all symbols. For example, in Fig. 1(a), what are "M-NCY-10", "M-NCY-15" and "M-NCY-30"? I only understand that M-NCY is MXene-based nanofibers coated yarns.

8) The authors reviewed many 2D materials, which are synthesized by the topochemical method. Thus, it might be helpful to make a table for these materials, including their energy applications, to summarize them.

Round 2

Reviewer 2 Report

The authors have adequately addressed my concerns and the manuscript is suitable for publication in Micromachines